# KOnezumi-AID: Automation Software for Efficient Multiplex Gene Knockout Using Target-AID

**DOI:** 10.3390/ijms252413500

**Published:** 2024-12-17

**Authors:** Taito Taki, Kento Morimoto, Seiya Mizuno, Akihiro Kuno

**Affiliations:** 1College of Biological Sciences, University of Tsukuba, Tsukuba 305-8572, Japan; s2110307@u.tsukuba.ac.jp; 2Doctoral Program in Medical Sciences, Graduate School of Comprehensive Human Sciences, University of Tsukuba, Tsukuba 305-8575, Japan; s2030401@u.tsukuba.ac.jp; 3Laboratory Animal Resource Center, Transborder Medical Research Center, Institute of Medicine, University of Tsukuba, Tsukuba 305-8575, Japan; 4Department of Anatomy and Embryology, Institute of Medicine, University of Tsukuba, Tsukuba 305-8575, Japan

**Keywords:** CRISPR, gene knockout, multiple gene knockout, genome editing, base editing, multiple base editing, Target-AID, bioinformatics, software

## Abstract

With the groundbreaking advancements in genome editing technologies, particularly CRISPR-Cas9, creating knockout mutants has become highly efficient. However, the CRISPR-Cas9 system introduces DNA double-strand breaks, increasing the risk of chromosomal rearrangements and posing a major obstacle to simultaneous multiple gene knockout. Base-editing systems, such as Target-AID, are safe alternatives for precise base modifications without requiring DNA double-strand breaks, serving as promising solutions for existing challenges. Nevertheless, the absence of adequate tools to support Target-AID-based gene knockout highlights the need for a comprehensive system to design guide RNAs (gRNAs) for the simultaneous knockout of multiple genes. Here, we aimed to develop KOnezumi-AID, a command-line tool for gRNA design for Target-AID-mediated genome editing. KOnezumi-AID facilitates gene knockout by inducing the premature termination codons or promoting exon skipping, thereby generating experiment-ready gRNA designs for mouse and human genomes. Additionally, KOnezumi-AID exhibits batch processing capacity, enabling rapid and precise gRNA design for large-scale genome editing, including CRISPR screening. In summary, KOnezumi-AID is an efficient and user-friendly tool for gRNA design, streamlining genome editing workflows and advancing gene knockout research.

## 1. Introduction

Since the establishment of the CRISPR-Cas9 genome-editing technology, the speed at which genetically modified mice are generated has markedly increased [1,2]. According to the International Mouse Phenotyping Consortium, until 2021, knockout (KO) mice have been used to identify approximately 11,241 protein-coding genes [3], accounting for nearly half of all known mouse genes, with this number continuously growing. Single-gene KO analyses could possibly be completed in the near future; establishing and analysing multiplex KO mutants is a critical challenge [4,5].

KO using the CRISPR-Cas9 system poses the risk of inducing large-scale chromosomal deletions spanning several thousand base pairs triggered by DNA double-strand breaks (DSBs) [6]. This risk increases significantly when multiple genes are simultaneously knocked out [7]. To address this issue, we focused on base editing using Target-AID, a synthetic complex of Cas9 derived from *Streptococcus pyogenes* (SpCas9) fused to activation-induced cytidine deaminase (AID), for gene editing without the need for DSBs. Target-AID is a base editor converting cytosine to thymine (C-to-T), thereby introducing point mutations 17 to 19 nucleotides (nt) upstream of the protospacer adjacent motif (PAM) sequence [8]. Compared to other base editors, Target-AID has a narrower editing window [9], exhibiting high precision and reduced risk of unintended mutations. Additionally, compared to other base editors causing off-target effects on RNA, Target-AID decreases this risk [10], making it a safer option for the simultaneous editing of multiple genes.

KO strategies using base editors either induce premature termination codons (PTCs) to truncate mRNA [11,12] or introduce frameshift mutations via splice site disruption [13,14]. However, guide RNAs (gRNAs) for these strategies need to be manually designed for each gene, posing a significant obstacle for the design of experiments. Therefore, this study aimed to develop KOnezumi-AID, a tool for the automatic design of experiment-ready gRNAs for Target-AID, simply by inputting the target genes to be knocked out. By allowing gene symbols as inputs, this tool facilitates the rapid design of genome editing strategies, including those for the KO of multiple genes.

## 2. Results

### 2.1. Implementation of KOnezumi-AID

We implemented a command-line tool, KOnezumi-AID, to design optimal gRNAs for KO of protein-coding genes in mice using SpCas9-based Target-AID. We focused on mice because they are suitable models for the rapid generation of KO mutants and have abundant publicly accessible genomic data [15,16]. KOnezumi-AID requires a refFlat file and a reference sequence as inputs for preprocessing. To extract gRNA candidates for protein-coding exons, KOnezumi-AID filters the refFlat file by removing the non-protein-coding transcripts and duplicate transcripts. Additionally, KOnezumi-AID targets both autosomes and sex chromosomes, excluding alternative assemblies (see Section 4 for details). Next, the sequences of transcripts in the transcriptional direction are extracted according to the genomic annotations of the filtered refFlat file and saved as refseq_to_transcribed_regions.pkl. To standardise the subsequent computational processes, genomic coordinates in each transcript are converted into relative coordinates starting from the transcribed regions using bedtools [17] and cached as unique_coding_refflat.pkl (Figure 1A).

After preprocessing, KOnezumi-AID accepts a gene symbol or RefSeq ID as input from the user and identifies the optimal gRNA candidates for knocking out the target gene. The gRNA design strategy of KOnezumi-AID focuses on inducing a PTC or causing a frameshift mutation by disrupting the splice acceptor or donor consensus nucleotides [11,12,13,14]. Experiment-ready gRNAs are printed to the standard output and saved as comma-separated value (CSV) files in the directory konezumiaid_data/output (Figure 1B).

For gRNA design, potential gRNA candidates were first comprehensively listed. As the target window for Target-AID is the cytosine +17 to +19 nt from the SpCas9-dependent PAM (NGG) [7], sequences containing both the targetable cytosine and PAM at the exon or splice sites were extracted. To induce a PTC, the target cytosine must be within the CAG, CGA, or CAA codons. For splice sites, we ensured that the PAM was located between +17 and +19 nt. To induce a PTC, codons were calculated using the relative position of the open reading frame, and candidate gRNAs resulting in a stop codon via a C-to-T conversion were identified. When targeting splice sites, the surrounding sequences at the exon start and end points were retrieved, and gRNAs targeting the nucleotide conversion of the splice donor or acceptor were identified (Figure 1C).

After listing all candidate gRNAs, KOnezumi-AID filtered them to identify experiment-ready gRNAs. In the first filtration step, gRNAs containing TTTT in the target sequence were excluded, as the use of pol III promoters for gRNA expression is a common vector-based approach, and the TTTT sequence triggers pol III termination [18]. Next, we established rules to extract gRNAs that are likely to induce KO. For single-exon genes, we selected gRNAs inducing PTCs within the first 50% of the coding sequence (CDS) as candidates [15]. For multi-exon genes, we prioritised the triggering of non-sense-mediated mRNA decay (NMD), an mRNA quality control mechanism that degrades transcripts with abnormal stop codons via PTC induction [19]. The gRNAs inducing PTCs in the last exon were excluded, as they do not lead to NMD [20,21]. Next, flags were assigned based on the following criteria [22]: start 150 nt rule, indicating that NMD is less likely if the PTC is within 150 nt of the start codon [22,23]; last 50 nt rule, indicating that NMD is less likely if the PTC is within 50 nt of the last exon junction complex [24]; and long exon rule, indicating that exons > 400 nt suppress NMD [25]. For gRNAs targeting splice sites, those affecting exons whose lengths were multiples of three were excluded because they do not induce frameshifts. Furthermore, gRNAs targeting the last exon were excluded to avoid the potential splicing of other genes. Finally, gRNAs targeting the second to last exon were excluded, as PTC induction via frameshifts in this region does not trigger NMD (Figure 1D).

The output of KOnezumi-AID includes the gRNA sequence with the PAM, the link to CRISPRdirect [26], and targeted amino acids for gRNAs designed to induce PTCs. Links to CRISPRdirect help users to efficiently select gRNAs with a minimal number of potential off-target sites. If a gene has multiple isoforms, gRNAs targeting all isoforms are included in the output (Figure 2A). For input genes with a single transcript or those specified by the RefSeq ID, the amino acid number of the target site is shown. Additionally, for multi-exon genes, the ‘Recommended’ column is included, marked as True when all four criteria for NMD susceptibility are met, indicating a high likelihood of NMD. For gRNAs inducing exon skipping by disrupting the splice sites, the position of the targeted exon is shown (Figure 2B). The execution time of KOnezumi-AID was less than 2 s for 18,536 (89.4%) mouse genes. Only 113 genes required more than 10 s, with the longest processing time being 144 s. The Pearson correlation between the execution time and the total number of sequences containing “C” within the target window was 0.87, indicating that the processing time for each gene is dependent on the number of such sequences (Figure 2C).

### 2.2. Gene Targetability Assessment Using KOnezumi-AID

The development of KOnezumi-AID enabled us to comprehensively assess whether gRNAs can be designed for KO using Target-AID. A total of 20,725 mouse gene symbols, corresponding to protein-coding genes included in the filtered refFlat (see details in Section 4), were searched using KOnezumi-AID, of which 17,233 genes (83.2%) were identified as targetable genes with at least one designable gRNA. Specifically, 16,814 genes (81.1%) had gRNAs capable of inducing PTCs, whereas 9984 genes (48.2%) could be targeted with gRNAs inducing exon skipping by disrupting splice sites (Figure 3A). An examination of the distribution of gRNAs per gene revealed that the largest group consisted of genes for which gRNA design was not possible. As the number of gRNAs increased, the number of target genes decreased. Yet, when calculating the total number of genes with 10 or more successfully designed gRNAs, this group accounted for the largest proportion (Figure 3B). Furthermore, an analysis of exon numbers revealed that, among the 2037 single-exon genes, 1213 were targetable by at least one gRNA, confirming that the KO strategy for single-exon genes was effective. Similarly, the high targetability of genes with 20 or more exons (2478 of 2516) highlighted the versatility of our approach across different gene structures (Figure 3C). Our analysis also revealed the characteristics of the non-targetable genes. These genes exhibited a short CDS (Appendix A). By summing the number of genes with up to five exons, this group accounted for 66.2% of all non-targetable genes (Appendix A). These results suggest that genes with a short CDS and few exons are more likely to be non-targetable. To further investigate the characteristics of non-targetable genes, several factors were examined. First, the design of gRNAs for inducing PTCs were analysed by comparing the proportions of arginine, glutamine, and tryptophan residues, which are targetable by Target-AID. After dividing the genes into non-targetable and targetable groups and excluding transcripts suspected of annotation errors, significant differences were observed in the proportions of arginine and glutamine between the two groups, while no significant difference was found for tryptophan (Appendix A). Additionally, the total number of sequences containing C within the target window was significantly lower in non-targetable genes (Appendix A). These findings suggest that non-targetable genes are characterized by a lower proportion of certain target amino acids and the absence of sequences targetable by Target-AID. Furthermore, for gRNAs targeting splice sites, we compared the number of splice sites that did not conform to the GT–AG rule and found no significant differences (Appendix A). This suggests that the inability to target splice sites is likely due to the presence of few exons or the absence of PAM sequences. Overall, these results confirm the effectiveness and flexibility of KOnezumi-AID in targeting various genes, including single- and multi-exon genes.

### 2.3. Application of KOnezumi-AID to Human Genome Data

The gRNA design strategy of KOnezumi-AID is not limited to mice and can be applied to various species, including animals, plants, and even microbiomes, provided that their genomes have annotated exons. Therefore, to verify the robustness of the KOnezumi-AID workflow, we evaluated the targetability, distribution of gRNAs per gene, and number of exons in genes where gRNAs were designed using human genes. Of the 19,073 human gene symbols analysed, 15,561 genes (81.6%) were targetable, with at least one designable gRNA identified using Target-AID. Specifically, 15,112 genes (79.2%) had gRNAs capable of inducing PTCs, whereas 9007 genes (47.2%) were targetable for exon skipping by disrupting the splice sites (Figure 4A). The characteristics of genes with designed gRNAs in humans mirrored those observed in mice (Figure 4B,C). These findings confirmed that the KOnezumi-AID workflow was applicable to human genome data.

### 2.4. Batch Processing Capacity of KOnezumi-AID

KOnezumi-AID has batch processing to streamline gRNA designs for large-scale CRISPR screening, which enables the simultaneous targeting of multiple genes [27]. The user prepares a CSV or Excel file devoid of headers containing the gene symbols or RefSeq IDs of interest. By executing KOnezumi-AID using this file, the users can retrieve the search results for multiple genes (Figure 5A). To evaluate the performance of batch processing, we investigated its execution time using random gene lists containing 10, 50, 100, 500, and 1000 genes. Those lists contain genes that are typically processed in under 2 s since approximately 90% of the genes are processed within this range. The results revealed that the execution time for batch processing increased linearly with the number of genes in various group combinations (Figure 5B). Notably, the batch processing time was shorter than the execution times of individual genes processed separately (Appendix A). Therefore, this batch processing functionality allows users to efficiently design gRNAs for numerous genes, enabling quick and accurate gRNA design for large-scale genome editing projects.

## 3. Discussion

Base editing is a recently developed genome editing technique combining CRISPR system components with additional enzymes to induce precise point mutations in DNA or RNA, thereby avoiding the need for DSBs. Specifically, Target-AID is widely recognised for its low risk of off-target RNA editing [10], thus facilitating the simultaneous editing of multiple genes.

In this study, we developed KOnezumi-AID, an automated tool for designing optimal gRNAs to induce KOs using Target-AID. Existing tools such as KOnezumi [28] and CHOPCHOP [29] can design gRNAs for gene KO. However, the strategy of these tools depends on DNA cleavage, which cannot exclude the possibility of causing unintended structural variations, especially when targeting multiple genes [6,7]. KOnezumi-AID is specifically designed to assist base editing with Target-AID, which can avoid the need for DSBs, offering enhanced support for KO of multiple genes.

Compared to other gRNA design tools for base editing, such as BE-Designer [30] and Benchling CRISPR Guide RNA Design Tool [31], which provide all possible base editor target sequences in a given input sequence, and BE-SCAN [32], which simplifies the selection of effective base editing tools to induce cancer-associated SNVs, KOnezumi-AID allows the users to design gRNAs for KO by simply inputting the target gene symbols or RefSeq IDs. Here, KOnezumi-AID targeted 83.2% of the genes in mice and 81.6% of the genes in humans. Its ease of use combined with its high targeting potential for Target-AID highlights KOnezumi-AID as a key tool for the efficient and reproducible design of gRNAs.

This study has several limitations. The most significant limitation is the user interface of KOnezumi-AID. Compared to the original KOnezumi tool, which offers a user-friendly web application [28], KOnezumi-AID is a command-line tool that can only be used by those familiar with terminal operations. Therefore, the implementation of a graphical user interface is an urgent task for KOnezumi-AID. Moreover, since KOnezumi-AID currently supports only Target-AID, leaving 17% of genes in mice without experiment-ready gRNA designs. Genes that cannot be targeted by Target-AID may be edited using other base editors or Target-AID-NG, which recognises NG PAM sequences instead of NGG [33], thus potentially enabling gRNA design for all genes. Finally, gRNAs reported by KOnezumi-AID lack decisive information regarding potential off-target effects and on-target efficiency, as the determinants of these factors in Target-AID remain unclear. In terms of off-target effects, KOnezumi-AID provides a link to CRISPRdirect, which reports gRNA matches to the target sequence (20-mer) and their seed sequence (12- or 8-mer). Since Target-AID is based on CRISPR-Cas9 [8], its off-target effects are expected to be similar to those of conventional Cas9. CRISPRdirect facilitates the selection of optimal gRNAs. Nonetheless, as the determinants of off- and on-target effects in Target-AID remain unclear, further research including experimental validations is required to enable more effective and precise gRNA design.

In the future, integrating base editors beyond Target-AID into KOnezumi-AID and expanding its applicability to other species will enhance its versatility. Notably, as nonsense mutations account for approximately 11% of all identified genetic alterations responsible for human inherited diseases [34], the use of base editing to generate such disease models through precise single-nucleotide mutations shows significant potential to advance the research on disease mechanisms. KOnezumi-AID serves as a powerful tool for such studies, particularly those on single-nucleotide variants related to human diseases.

## 4. Materials and Methods

### 4.1. Datasets

All genomic data used in this study were downloaded from the UCSC Genome Browser [16]. Assembled versions of the reference genome sequence and refFlat genome annotation for mice and humans were GRCm39/mm39 (GCA_000001635.9) and GRCh38/hg38 (GCA_000001405.29), respectively. KOnezumi-AID defines non-protein-coding transcripts as those with RefSeq ID starting with ‘NR’, duplicated transcripts with same ‘NM’ number for a single gene symbol, and alternative assemblies as chromosomes including ‘_alt’, ‘_random’, ‘_fix’, and ‘_Un’ suffix.

### 4.2. Targetable Splice Sites

KOnezumi-AID defines splice sites according to the conventional GT–AG rule as those flanked by GT bases at the 5′ end and AG bases at the 3′ end [35]. KOnezumi-AID searches for gRNAs including the G of the splice site, which can be edited by Target-AID on the reverse strand (G-to-A conversion) within the target window PAM (CCN) −17 to −19 nt.

### 4.3. Measurement of Execution Time

Since KOnezumi-AID can be executed on Windows systems after preprocessing, the execution time was measured on a machine equipped with an Intel i5-13600K processor and 128 GB of RAM, using PowerShell 7.4.6. The execution time was defined as the duration from the start time to the end time of the command. Genes used in the batch process were randomly selected using the random module from the standard Python library.

### 4.4. Statistical Analysis

A two-sided Welch’s *t*-test was performed using the SciPy version 1.12.0 to compare the CDS lengths of KOnezumi-AID targetable and non-targetable genes.

### 4.5. Code Availability

Operation of KOnezumi-AID was validated on WSL2 with Ubuntu as well as on macOS using Python versions 3.9–3.11. In this study, KOnezumi-AID version 0.3.6.1 was used (https://dx.doi.org/10.5281/zenodo.14402527, accessed on 15 December 2024). KOnezumi-AID is available as an open-source software tool under the MIT licence at https://github.com/aki2274/KOnezumi-AID, accessed on 15 December 2024. The PyPI package is available at https://pypi.org/project/KOnezumiAID/, accessed on 15 December 2024.

## Figures and Tables

**Figure 1 ijms-25-13500-f001:**
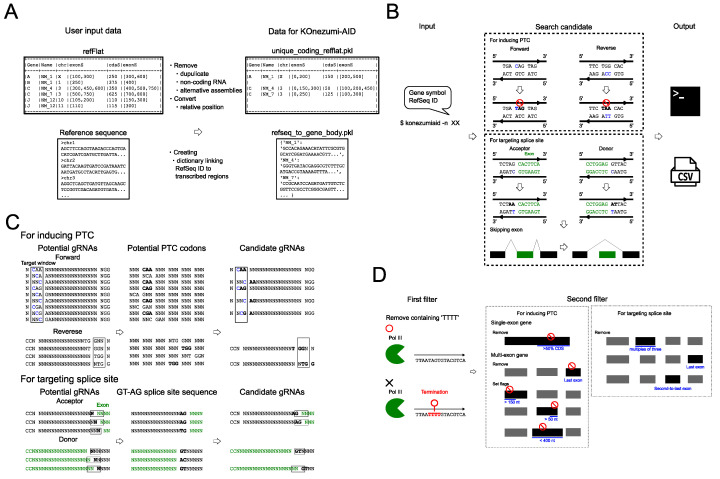
Overview of KOnezumi-AID. (**A**) Preprocessing in KOnezumi-AID. A refFlat file and a reference sequence are processed into a format suitable for KOnezumi-AID search. (**B**) Main search process of KOnezumi-AID. Editable bases are indicated in blue. Bold and red stop marks indicate the premature termination codons (PTCs). Exons are marked in green. (**C**) Search process of KOnezumi-AID shown in a schema. Black box indicates the target window. Editable bases are indicated in blue. Regions where codons can be modified to introduce PTCs or target the GT–AG splice site consensus sequence are highlighted in bold. Exons are marked in green. (**D**) Filtering process in KOnezumi-AID. Red stop marks indicate the PTCs. Key filtering checkpoints are indicated in blue.

**Figure 2 ijms-25-13500-f002:**
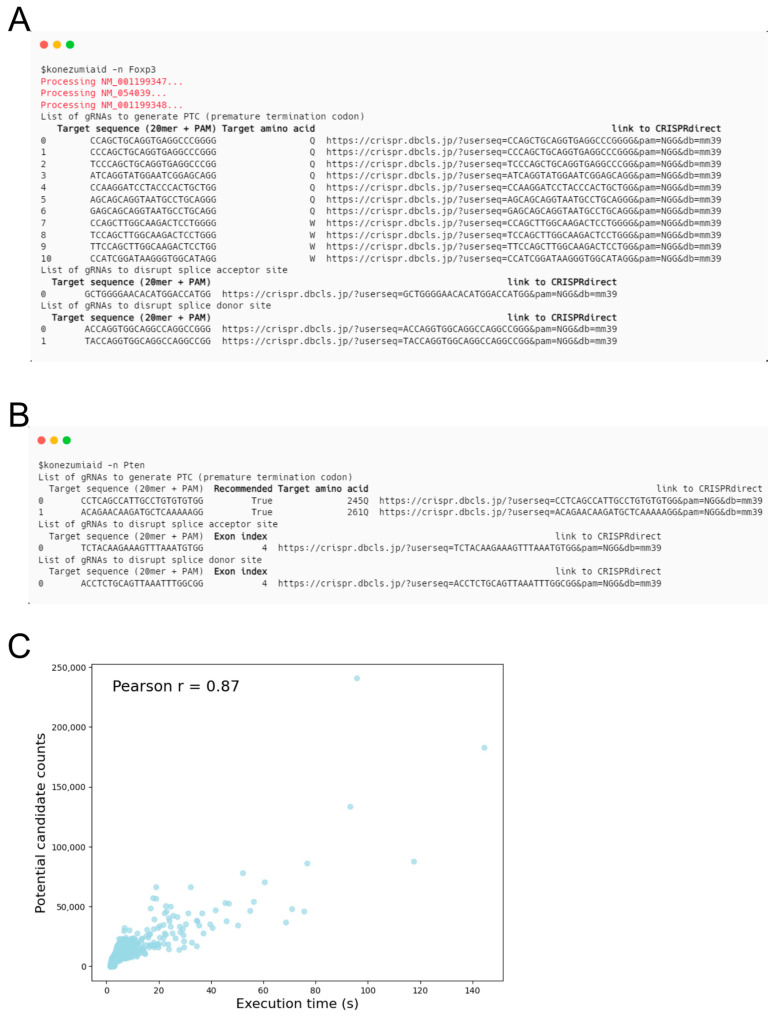
Output of KOnezumi-AID. (**A**) Output of KOnezumi-AID for a multi-isoform gene. The output shows guide RNAs (gRNAs) inducing PTCs and those disrupting the acceptor or donor consensus nucleotides for genes with multiple isoforms. Multiple isoforms being searched simultaneously are indicated in red. Output columns are highlighted in bold. (**B**) Output of KOnezumi-AID for a single-isoform gene. The output shows gRNAs inducing PTCs and those disrupting the acceptor or donor consensus nucleotides for a gene with a single isoform. Columns that are specifically displayed when searching for single-isoform genes are highlighted in bold. (**C**) Scatter plot shows the relationship between the execution time (in seconds) and the total number of sequences containing “C” within the target window for each gene. The Pearson correlation coefficient is 0.87.

**Figure 3 ijms-25-13500-f003:**
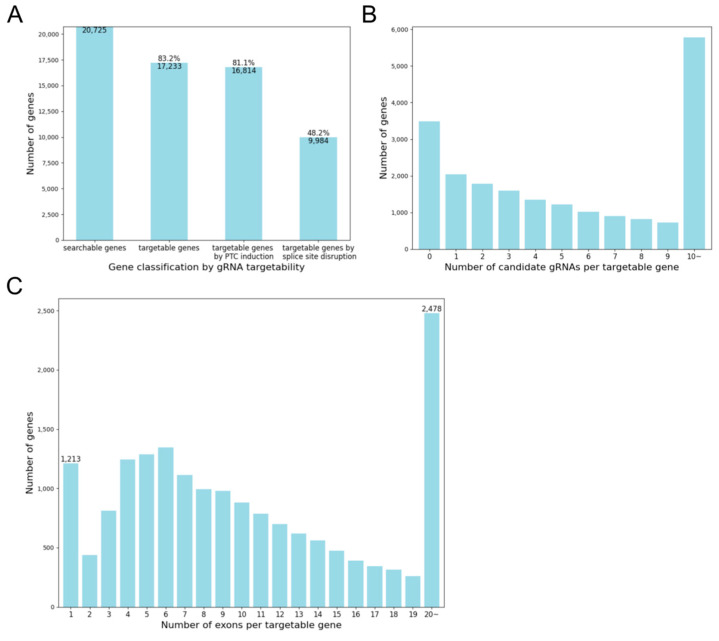
Analysis of candidate gRNA-associated mouse genes and their distribution. (**A**) Number of genes with gRNA targetability. The total number of analysed genes (searchable genes), number of genes with at least one candidate gRNA (targetable genes), number of genes with candidate gRNAs inducing PTCs (targetable genes by PTC induction), and number of genes with candidate gRNAs disrupting splice sites (targetable genes by splice site disruption) are displayed. (**B**) Number of genes by number of candidate gRNAs. Distribution of genes based on the total number of candidate gRNAs, including those inducing PTC and those disrupting splice sites, is shown. For visualization, counts are capped at 10, even if the number of gRNAs per gene exceeds 10. (**C**) Number of genes by number of exons. Distribution of exons for genes targetable by gRNAs is shown. For visualization, counts are capped at 20, even for genes with more than 20 exons.

**Figure 4 ijms-25-13500-f004:**
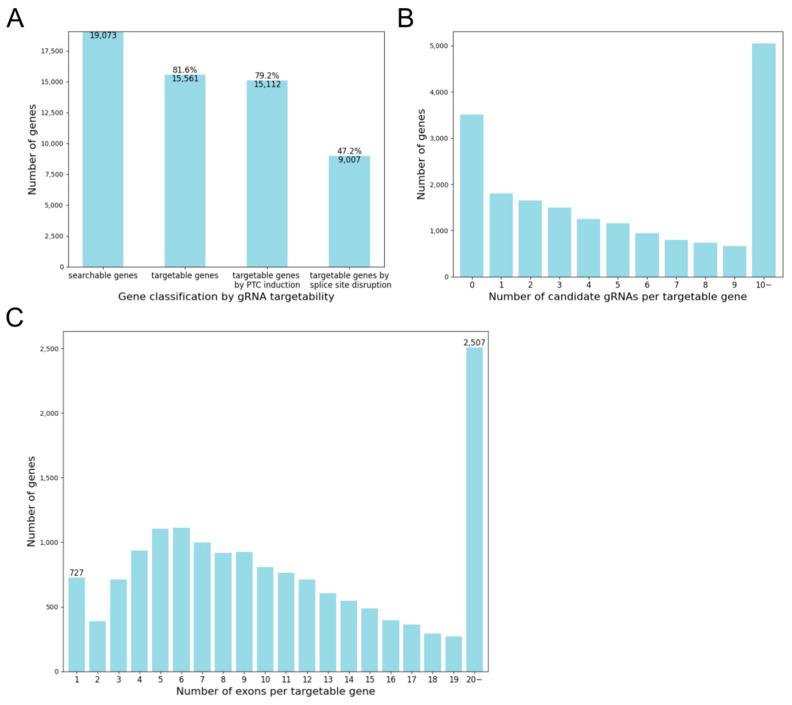
Application of KOnezumi-AID to human gene data. (**A**) Number of genes with gRNA targetability. The total number of analysed genes (searchable genes), number of genes with at least one candidate gRNA (targetable genes), number of genes with candidate gRNAs inducing PTCs (targetable genes by PTC induction), and number of genes with candidate gRNAs disrupting splice sites (targetable genes by splice site disruption) are displayed. (**B**) Number of genes by number of candidate gRNAs. Distribution of genes based on the total number of candidate gRNAs, including those inducing PTCs and those disrupting splice sites, is shown. For visualization, counts are capped at 10, even if the number of gRNAs per gene exceeds 10. (**C**) Number of genes by number of exons. Distribution of exons for genes targetable by gRNAs is shown. For visualization, counts are capped at 20, even for genes with more than 20 exons.

**Figure 5 ijms-25-13500-f005:**
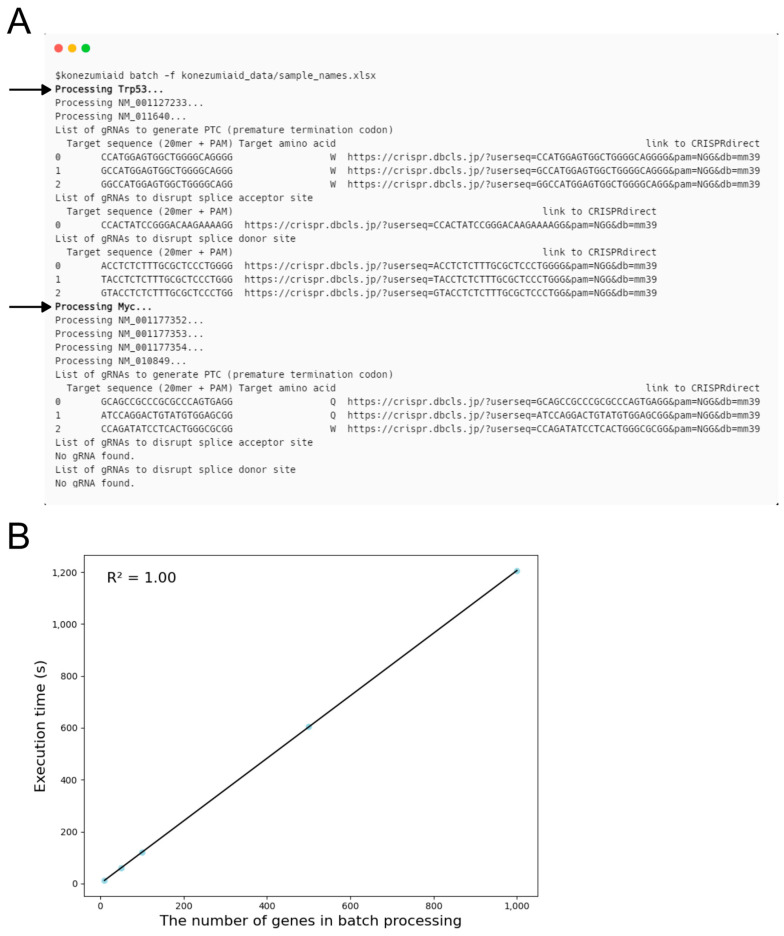
Batch processing. (**A**) Output of KOnezumi-AID for batch processing. The results of batch searches using an Excel file containing the gene abbreviations, *Trp53* and *Myc*, are shown. Two genes being searched sequentially are indicated in bold by arrows. (**B**) Scatter plot illustrates the relationship between the number of genes in batch processing and execution time (in seconds). Light blue dots represent the average execution time for each batch processing epoch, while black line shows the linear regression line. The coefficient of determination (R^2^) is 1.

## Data Availability

Data generated or analysed in this study are available upon request from the corresponding authors.

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
