# Peer review of "KOnezumi-AID: Automation Software for Efficient Multiplex Gene Knockout Using Target-AID"

_ijms, 2024, doi:10.3390/ijms252413500_

Round 1

Reviewer 1 Report

Comments and Suggestions for Authors

Taki, et. al. presents KOnezumi-AID, a command-line software designed to optimize gRNA design for Target-AID-mediated genome editing. The tool supports batch processing for high-throughput applications and is validated on mouse and human genomes, demonstrating its utility in large-scale genome editing projects. However, several key questions remain that should be properly addressed before it can be accepted.

1.     While the manuscript emphasizes the precision of Target-AID, it lacks an in-depth discussion of potential off-target effects, which including experimental validation or detailed analysis of off-target editing rates. It will strengthen the reliability of the claims.

2.     The manuscript briefly mentions other gRNA design tools, such as BE-Designer, BE-SCAN and KOnezumi, especially for KOnezumi, which offers a user-friendly web application. Can authors compare the difference between KOnezumi-AID and KOnezumiBE/Designer/BE-SCAN? What is the advantage of KOnezumi-AID?

3.     The authors showed KOnezumi-AID can target 86.4% of the genes in mice and 85.5% of the genes in humans. Could authors explain why the left 15% genes cannot be targeted? In addition to Target-AID, is there other experimental methods can be included to further improve the usage scenario of KOnezumi-AID?

4.     The running time for KOnezumi-AID is not specified. Providing performance for the time required to process a single gene or a batch of genes would help assess its efficiency.

Reviewer 2 Report

Comments and Suggestions for Authors

This manuscript is well written and a pleasure to read. The authors presented a useful tool to design guide RNAs for Target-AID. It supports the design of gRNAs for the simultaneous knockout of multiple genes and allows batch processing. This is particularly useful for large-scale genome editing projects. The authors demonstrated KOnezumi-AID can effectively design gRNAs in mouse and human.

Although KOnezumi-AID is already powerful, it has several limitations. As the authors mentioned in their paper, lack of a web-based intuitive user interface limits the accessibility for users at all levels. Since Target-AID is particularly promising for applications in microbiome therapeutics, expending KOnezumi-AID’s capability to support other species is needed. I’m glad the authors put these in their future development.

I’m aware that CHOPCHOP and Benchling are also capable of designing gRNAs for multiple genes simultaneously. CHOPCHOP supports multiple CRISPR-based applications, including base editing, and various species, and it provides a user-friendly interface. I’d like to invite the authors to compare KOnezumi-AID with other tools and explain why we should choose KOnezumi-AID over others, like CHOPCHOP.

Round 2

Reviewer 1 Report

Comments and Suggestions for Authors

The authors have answered all my question.